

# Evolution of gene regulatory networks by means of selection and random genetic drift

Stefanos Papadadonakis[1,2], Antonios Kioukis[3], Charikleia Karageorgiou[4] and Pavlos Pavlidis[1,2]

[1] Institute of Computer Science, Foundation for Research and Technology Hellas, Heraklion, Crete, Greece
[2] Department of Biology, University of Crete, Heraklion, Crete, Greece
[3] School of Medicine, University of Crete, Heraklion, Crete, Greece
[4] Department of Biological Sciences, University at Buffalo, New York, New York, United States

## ABSTRACT

The evolution of a population by means of genetic drift and natural selection operating on a gene regulatory network (GRN) of an individual has not been scrutinized in depth. Thus, the relative importance of various evolutionary forces and processes on shaping genetic variability in GRNs is understudied. In this study, we implemented a simulation framework, called EvoNET, that simulates forward-in-time the evolution of GRNs in a population. The fitness effect of mutations is not constant, rather fitness of each individual is evaluated on the phenotypic level, by measuring its distance from an optimal phenotype. Each individual goes through a maturation period, where its GRN may reach an equilibrium, thus deciding its phenotype. Afterwards, individuals compete to produce the next generation.
We examine properties of the GRN evolution, such as robustness against the deleterious effect of mutations and the role of genetic drift. We are able to confirm previous hypotheses regarding the effect of mutations and we provide new insights on the interplay between random genetic drift and natural selection.

# INTRODUCTION

The path from genotype to phenotype is characterized by an immense number of direct and indirect gene interactions. The relationship between genotype and phenotype has long been of interest to geneticists, developmental biologists and evolutionary biologists. This is partially because the relationship between them is ambiguous and non-linearities appear often. The same phenotype can manifest through a multitude of genetic variations a phenomenon often referred to as phenotypic plasticity. Conversely, a singular genetic makeup has the potential to yield diverse phenotypic outcomes, as it interacts with varying environmental conditions (*Sansom & Brandon, 2007*). Population genetics processes such as natural selection and random genetic drift operate on various levels of genomic organization, from single nucleotides, genes, networks of genes to complex phenotypes. Phenotypic variation may be directly affected by mutations but also by the interaction of

Corresponding author
Pavlos Pavlidis,
pavlos.pavlidis@uoc.gr

mutations at the same or different genes. As *Lehner (2007)* points out, 'Probably all heritable traits, including disease susceptibility, are affected by interactions between mutations in multiple genes'. Thus, it may seem incomplete that neutrality tests for the localization of natural selection, use solely genotypic information in models that incorporate no gene interactions or genotypic-phenotypic relations. In particular, selective sweeps, the process where a beneficial genetic mutation quickly spreads through a population, utilize the concept of constant selection coefficient, which can be understood as a summary of the dynamics of the allele under selection, but lacks a clear biological meaning (*Chevin & Hospital, 2008*). The concept of selective sweep theory is attractive for its straightforwardness, allowing researchers to develop software capable of identifying and pinpointing genomic areas potentially harboring mutations subject to natural selection. Consequently, selective sweep software are utilized to investigate whether a gene underwent recent and intense selection pressure, although they overlook the possibility that natural selection might occur through mechanisms diverging from a conventional selective sweep. If a genomic region is identified as the target of positive selection, the next step usually comprises an extensive literature search in an effort to connect the genotype to phenotype, and thus build plausible narratives that explain the action of positive selection (*Pavlidis et al., 2012*). Yet, if natural selection does not exclusively act on discrete mutations, identifying the targets of selection becomes challenging due to the (probably slower or even competing) dynamics of beneficial genotypes. *Chevin & Hospital (2008)* extended the theory of positive selection to the context of loci that affect a quantitative trait, that harbors background genetic variation due to other, unlinked, non-interacting loci. They assumed a large number of background loci with small effect on the phenotype. Even though the increase in frequency of a beneficial mutation is slower than the classical one-locus selective sweep, they showed that under such a model, the signature of a selective sweeps can still be detected at the focal locus, especially if the genetic variation of the background is limited. *Pavlidis, Metzler & Stephan (2012)* showed that when the trait under selection is controlled by only a few loci (up to eight in their simulations), it is possible that an equilibrium is reached, resulting in no fixation of a specific allele. Such equilibrium scenario occurs more frequently when loci have a similar effect on the phenotype. Contrariwise, if the population is far from the optimum and the focal allele has a relatively large effect, then it will reach fixation. In general, multi-locus models allow competition between loci, thus the time of a potential fixation of the selected allele(s) depends crucially on the initial conditions affecting whether a selective sweep will appear. This problem is even more pronounced when the phenotype in question is controlled by a gene-regulatory network, where the expression of a gene is affected by interactions between multiple genes.

To our knowledge, the first attempt to understand the evolution of regulatory networks was done in the seminal work by *Wagner (1996)*. Wagner formulated the numerical evolution a network of genes that assumed binary states (either expressed or not expressed). He studied whether a population of such networks can mitigate (buffer) the (detrimental) effect of mutations after it evolves to reach its optimum. Indeed, he found that after evolving a network of genes by means of natural selection (stabilizing selection),

the effect of mutations is considerably lower than a system where evolution has not occurred yet. Natural selection, combined with neutral processes, modifies gene expression and in consequence the properties of GRNs. *Ofria, Adami & Collier (2003)*, using computer simulations, demonstrated that when mutation rate is present, selection favors GRN variants that have similar phenotypes. *Wagner (2008)* showed that neutral variants with no effect on the phenotype facilitate evolutionary innovation because they allow for thorough exploration of the genotype space. These ideas can be directly applied to GRNs by employing the concepts of robustness and redundancy. Robustness refers to the resilience that GRNs exhibit with respect to mutations. One mechanism for maintaining robustness is redundancy. Redundancy may be caused by/implemented by gene duplication or by unrelated genes that perform similar functions (*Nowak et al., 1997*).

Three deviations from classic selective sweep theory are possible because of positive selection effects on GRNs: i) variation in selection intensity through time; ii) 'soft' sweeps that start with several favorable alleles; and iii) overlapping sweeps (*Hermisson & Pennings, 2005*). Since more than one network configuration can give rise to the same phenotype, the patterns of polymorphisms at the genome level are not necessarily expected to follow distributions similar to ones that arise by a strong beneficial mutation in just a single gene (*Pavlidis, Metzler & Stephan, 2012*). Adaptation may often be based on pre-existing genetic variation of the population (standing genetic variation), rather than single, new mutations. Thus, it is expected that the selected allele was once neutral standing variation, which will in turn weaken the signal of positive selection (*Przeworski, Coop & Wall, 2005*). Finally, if hitchhiking dominates the pattern of neutral diversity, the genome may be subject to multiple overlapping sweeps.

In this work, we study the evolution of a population of GRNs by means of random genetic drift and selection. For this reason we developed a forward-in-time simulator, named EvoNET that extends Wagner's classical model (*Wagner, 1996*) and subsequent extensions (*e.g.*, *Siegal & Bergman, 2002*) by (*i*) explicitly implementing *cis* and *trans* regulatory regions. *cis* and *trans* regions may mutate and interact, thus, affecting gene interactions and gene expression levels. In contrast, Wagner's model directly modifies the values of the interaction matrix without implementing any mutation model. In addition, (*ii*) we allow for viable cyclic equilibria during the maturation period in contrast to Wagner's model, where cyclic equilibria are considered lethal. We assume that such cyclic equilibria resemble circadian regulatory or expression alternations. Futhermore, (*iii*) we devised a different recombination model, where a set of genes with their *cis* and *trans* regulatory regions, can recombine in another background, with the subsequent consequences on their interactions with other genes. We provide results about the robustness of the network to mutations, and its properties during the traversal of fitness landscape. Portions of the Methods and Results sections were published as part of a preprint (*Kioukis & Pavlidis, 2019*).

## METHODS

### The model

#### *Regulatory regions define interactions*

We assume a population of $N$ haploid individuals. Individuals may have either a single parent or two parents. In the later case recombination is allowed (see "Inheritance of regulation and recombination"). Each individual comprises a set of $n$ genes consisting of *cis* and *trans* binary regulatory regions, each of length $L$. A *cis* regulatory region is defined as the region upstream the gene on which the *trans* regions of other genes of the GRN can bind. Let $R_{i,c}$ be the *cis* region of the gene $i$ and $R_{j,t}$ the *trans* region of gene $j$. Then, we define a function $I(R_{i,c}, R_{j,t})$ that receives as arguments two binary vectors and returns a float number in the $[-1, 1]$ representing the interaction type and strength. Negative values model suppression, positive values activation, and 0 means no interaction. Here, for the absolute value of interaction, we use Eq. (1):

$$|I(R_{i,c}, R_{j,t})| = \begin{cases} \dfrac{pc(R_{i,c}[1:L-1] \& R_{j,t}[1:L-1])}{L} \\ 0 : \text{no regulation} \end{cases} \tag{1}$$

where $pc$ is the popcount function, which counts the number of set bits (*i.e.*, 1's) that are common in the two vectors. The occurrence of interaction, as well as, the type (suppression or activation) is defined by the last bit of the $R_{i,c}$ and $R_{j,t}$ vectors as:

$$\begin{matrix} 0, & R_{i,c}[L] = 0 \\ +, & R_{i,c}[L] = R_{j,t}[L] = 1 \\ -, & R_{i,c}[L] = 1 \text{ and } R_{j,t}[L] = 0 \end{matrix} \tag{2}$$

In other words, the first $L-1$ bits define the strength of the interaction, which is proportional to the number of common set bits (*i.e.*, common 1's). The last ($L^{th}$) bit in each vector determines if the interaction is present and if it is suppression or activation. If the last bit of the *cis* element is '0' then it does not 'accept' any regulation. If it is '1', then regulation can be either positive or negative, depending on the last bit of the *trans* element.

The above representation of regulation enables a more realistic representation than Wagner's model (*Wagner, 1996*) and its more recent extensions (*Siegal & Bergman, 2002*; *Huerta-Sanchez & Durrett, 2007*). A single mutation in the *cis* region of a gene can affect its regulation by all other genes, and a mutation in the *trans* region of a gene can affect the way it regulates all other genes (see also 'Mutation model of regulatory regions').

#### *Interaction matrix and expression levels*

Interaction values of each individual are stored in a square $M_{n \times n}$ matrix of real values in the $[-1, 1]$ range, where $n$ is the number of genes in the network. A positive $M_{ij}$ value indicates that gene $j$ activates gene $i$, a negative value indicates suppression and 0 represents no interaction. Thus, the row $M_{i.}$ represents the interaction between all *trans* regulatory elements and the *cis* regulatory region of gene $i$. Gene expressions are represented by a vector $E_n$ of $n$ elements. In the general case, the expression level $E_j$ of the $j_{th}$ gene can be a real positive number. Here, however, $E$ is a binary vector, indicating only if

a gene is active or not. Such a representation is more efficient computationally. A similar approach has been used by *Wagner (1996)* and *Siegal & Bergman (2002)*.

### Inheritance of regulation and recombination

Each child inherits from its parents the *cis* and *trans* regulatory regions (the model allows for two parents or a single mother). The initial values of expression levels (at birth) are initialized to a constant binary vector. If the model allows for two parents, then recombination is possible. We have implemented two recombination models. The first is similar to *Wagner*'s *(1996)* model that swaps rows of the interaction matrix of parents to form children. Such a model results effectively in exchanging *cis* regulatory elements. Wagner's model of recombination may be, however, unrealistic because it allows only *cis* regulatory regions to be exchanged while *trans* regions do not recombine (Fig. 1, top panel). In *Wagner (1996)*, the interaction values between genes in the recipient and donor genomes remain unchanged after recombination (Fig. 1, upper panel A). We implemented Wagner's model of recombination, but we re-estimated the interaction values between genes in the donor and the recipient genomes. This is necessary because *cis* and *trans* interactions are modified after recombination (Fig. 1, upper panel B). We implemented an additional recombination model that allows cross-over events between parental genomes as follows: Assuming that the GRN consists of $n$ genes, let $j$, $0 < j < n$ be an recombination breakpoint. Then, the first $j$ genes inherit the *cis* and the *trans* regions from one parent, and the last $n - j$ genes inherit *cis* and *trans* regions from the other parent. The interactions between the first $j$ and the last $n - j$ genes are then re-computed in accordance to the resulting genome's regulatory regions (Fig. 1, bottom panel).

### Mutations

Mutations take place in the *cis* and *trans* regulatory regions during offspring generation. Since regulatory regions are implemented as binary vectors, a mutation can change a position in a region by modifying a 0 to 1 and *vice versa*. On one hand, if a mutation will affect a *cis* region, then all interactions between this *cis* and all *trans* regions might be modified (*i.e.*, the row of the interaction matrix will be affected). On the other hand, if a mutation will change a *trans* region, all interactions between this *trans* and all other *cis* regions might be modified (*i.e.*, the column of the interaction matrix). For each individual, the number of mutations is drawn from a Poisson distribution with parameter $\mu$ (mutation rate per genome per generation), and then mutations (if any) are placed uniformly among the *cis* and *trans* regulatory regions.

For example, let $R_{i,cis}$ be the *cis* regulatory region of gene $i$ that is going to be mutated. $R_{i,cis}$ comprises two parts: the $[1 : L − 1]$ part, which controls the strength of interactions and the $L$ position that controls the type of interaction as described in *Regulatory regions define interactions*. Since mutations in the $L$ position may have a dramatic effect, changing the type of interaction (*e.g.*, a repressor might become activator or regulation can be silenced), we implemented two different mutation rates for these two parts of the regulatory regions. Mutations in the first $[1 : L − 1]$ part are distributed uniformly. We model with 1% chance the probability that if a mutation occurs, the trans region
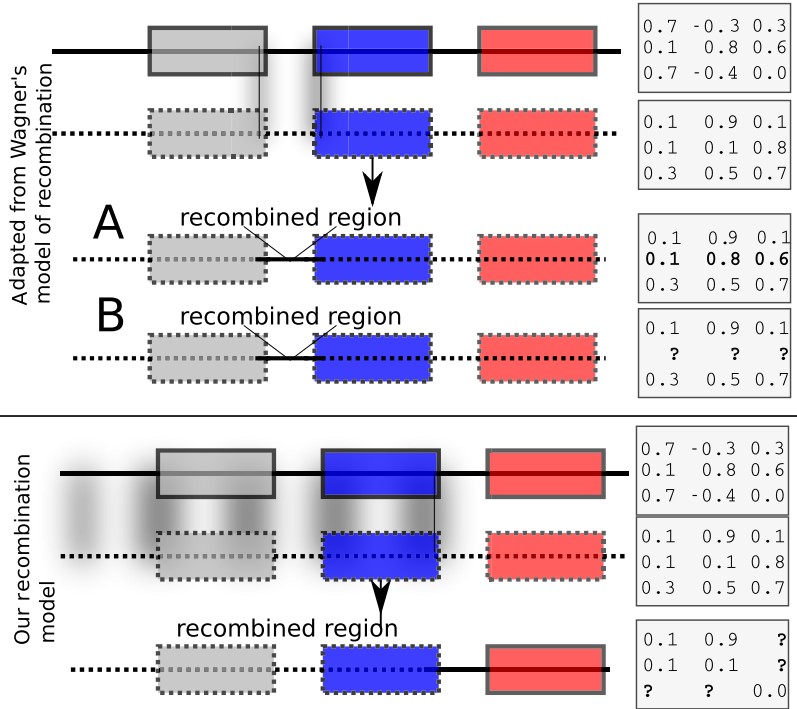

**Figure 1 Recombination models implemented by EvoNET.** Shaded areas show the gnomic regions that are exchanged due to the recombination process. At the upper panel, Wagner's model is illustrated, where cis regulatory regions can be swapped between individuals of the population. At the bottom panel, our model is shown. In our model, recombination is implemented via a recombination break-point. All genes at its left side inherit both the cis and the trans regions from one parent, whereas the genes on the right inherit cis and trans regions from the other parent. The interaction matrix is re-evaluated after recombination.

changes its behavior. This models the biological fact that mutations that change the nature of an established relationship of two genes are very rare as opposed to changing the strength of the respective relationship.

### Selection

The fitness of each individual is ultimately determined by their gene expression profile. Let $E_{opt}$ represent the optimal vector of expression values of the GRN. In EvoNET the user can opt to specify an optimal vector through the command line. The fitness of an individual $i$ with expression values defined by the $E_n^i$ vector is defined by:

$$F(E_n) = e^{-\left\|E_n^i - E_{opt}\right\|/\sigma^2} \tag{3}$$

where $\left\|E_n - E_{opt}\right\|$ is a norm of the difference between $E_n$ and $E_{opt}$ expression vectors (here the Euclidean distance is used). $\sigma^2$ is identical to the parameter $s$ of *Wagner (1996)*. This parameter models the 'strength of selection', *i.e.*, how pronounced is the effect of the differences in expression vectors to individuals' fitness. Parents are chosen proportionally to their fitness value $F(E_n^i)$.

*Maturation and equilibria*

Every 'new-born individual' has inherited the regulatory regions from its parents and by extension the interaction matrix (potentially with mutations) and has acquired an initial expression vector. Since genes may interact with each other, we have implemented an additional 'maturation' process during which, the expression levels of genes change, as a result of gene-gene interactions until either an equilibrium point, or a cyclic equilibrium is reached. At the $t+1$ step of the process a new expression vector $E_n(t+1)$ is obtained using the expression vector of the $t_{th}$ step and the interaction matrix $M$:

$$E_n(t+1) = ME_n(t). \tag{4}$$

Equivalently, the $i^{th}$ element $E_n(t+1)[i] = \sum_{j=1}^{n} M_{i,j}E_n(t)[j]$. Depending on the interaction matrix $M$ and the initial value of the expression vector $E_n$, there are three possible outcomes of this process.

$$
\begin{array}{lll}
(i) & E_n(t) = E_n(t+1) = E_n(t+2) = \dots \\
(ii) & E_n(t) = E_n(t+k) = E_n(t+2k) = \dots, & k > 1 \\
(iii) & E_n(t) \neq E_n(t+j), \forall \, t, j
\end{array} \tag{5}
$$

In Wagner's model (*Wagner, 1996*) as well as in *Huerta-Sanchez & Durrett (2007)*, only case $(i)$ in Eq. (5) is considered viable. Case $(i)$ facilitates fitness evaluation of the individual using Eq. (3). Individuals with a maturation process that concludes in $(ii)$ or $(iii)$ were removed from the population. Here, motivated by *Pinho, Borenstein & Feldman (2012)* who suggested that in Wagner's model most networks are cyclic, we developed a circadian framework to evaluate the fitness of individuals whose network maturation results in a cyclic equilibrium. Individuals that conclude in case $(iii)$, or individuals that conclude in case $(ii)$ but the period $k$ is greater than an upper threshold (defined as 10.000 steps in our simulations) were considered non-viable and were assigned a fitness of 0. If the maturation process concludes in case $(ii)$, with $E_n(t) = E_n(t+k) = E_n(t+2k) = \dots$ and $k < 10.000$, we evaluated the fitness of the individual as the minimum fitness value during the period of a cycle.

# RESULTS

## Comparisons between neutral evolution and selection scenarios
### Simulations setup

To explore the gene expression differences between neutral evolution and evolution under directional selection, we simulated neutral datasets and datasets under selection. All examples are provided in the Supplementary information. Both models were evolved for 15,000 generations. Each individual network comprises 10 genes, each with 30-bit long *cis* and *trans* regulatory elements. The last bit of each regulatory element is responsible for the type of regulation (positive or negative; see Methods) and the remaining 29 bits determine the strength of the interaction. In generation 0, all *cis*-regulatory elements were set so that they can not accept any regulation. In contrast, all *trans*-elements were set to be activators, thus they can regulate a *cis* element positively (provided that the last bit of the *cis*-element is 1). After maturation (see Methods), the expression vector was converted to binary

format (the expression value is 1 if the expression is positive and 0 otherwise). Thus, initially all expression vectors $v$ were equal to 0. The fitness of each individual was evaluated after maturation. The optimum was set to the state were all genes were expressed (*i.e.*, state 1 for all genes). For the simulations with selection, the selection intensity $1/\sigma^2$ (see Methods) was set to 1/5. The population size was set to 100 haploid individuals and remained constant throughout the entire simulation. Mutation rate was set to 0.005 unless stated otherwise.

### Optimum is gradually reached in a ladder-like fashion

We evaluated whether, and how, the population reaches the optimum state. Given that the initial state was 00000000 (*i.e.*, all genes inactive) and the optimum state was 11111111 (*i.e.*, all genes active), the population had to experience the appropriate changes in its *cis*- and *trans*- regulatory elements, and consequently the GRN, to achieve the activation of all genes. When mutation and recombination rates were sufficiently low, we observed a ladder-like behavior for the average fitness (Fig. 2); that is, networks were successively replaced by fitter networks in discrete steps.

At every step of the 'ladder', the average population fitness remains approximately constant. After reaching each fitness step, the population starts exploring different GRN topologies until a fitter genotype establishes in the population. While exploring candidate topologies, genetic drift acts and it is therefore possible that the population will not incorporate every novel beneficial network topology that it will encounter. If a beneficial topology overcomes drift, its frequency increases and the population average follows. Finally, when the new topology reaches fixation, the population has reached the next step in the fitness 'ladder' (Fig. 3).

Mutations and recombination are the driving force behind the exploration of the topology space, since they may result in a novel network topology. By increasing the mutation rate, the number of novel explored topologies increases and the time between each step decreases (Fig. S1). Recombination rates also affect the time required for each step. Recombination allows the parental networks to be combined resulting in enhancement of the network variability in the population, thus the optimum can be reached faster. In our simulations our proposed model R1R2 swapping reaches optimum faster than the row-swapping model proposed by *Wagner (1996)* (Fig. S4).

### Size of the regulatory space in neutrality and selection

We assessed how the population explores the state space of regulatory networks during its evolution, by evaluating the number of different genotypes present throughout the run. We studied whether neutrality or selection explores the space more efficiently, *i.e.*, which of the two processes allow the population to explore a higher number of genotypes on average. Under neutrality the genotype frequency was affected solely by genetic drift. In the limited amount of generations (15,000), and due to the small population size (100 individuals) the population explored a small fraction of the genetic landscape centered around the initial state. Namely populations on average harbored 5,105 distinct GRNs over the course of the simulation. In contrast, for scenarios involving selection, populations

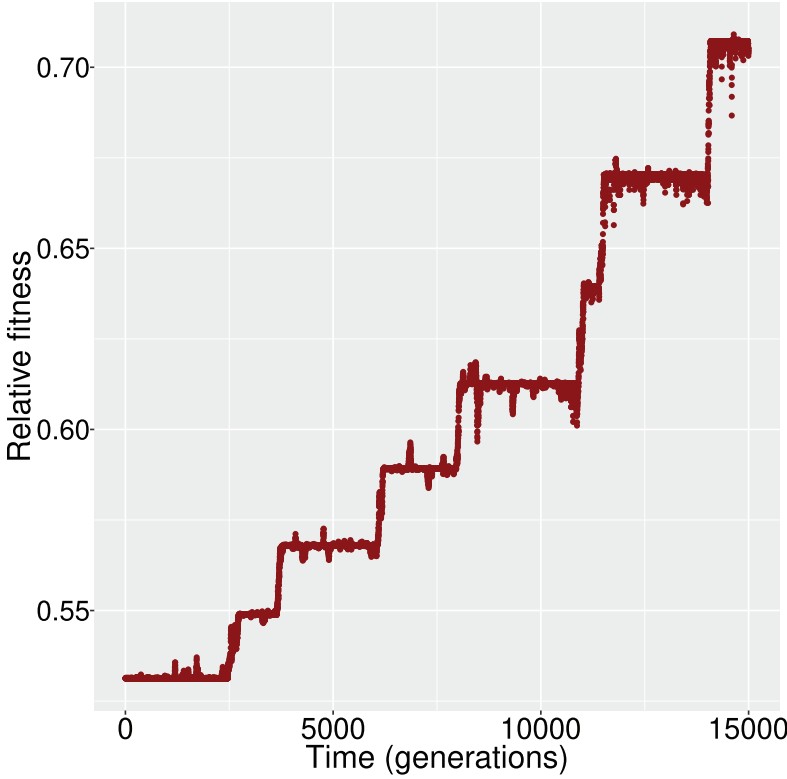

**Figure 2 The increment in average fitness of the population is taking place in discrete steps, in a ladder-like fashion.** This figure is one such example that demonstrates the fitness trajectory of the population.

encountered on average 17,110 distinct networks. We illustrate that the interplay between selection and drift is vital in this increase. After initialization, selection drives genotypes to local optima. It is plausible that more than one genotypes with similar fitness values are simultaneously present in the population at different frequencies (Fig. 3). Subsequently, neighbouring genotypes of similar fitness are explored solely by drift until a fitter one is found, whose frequency is increased and eventually it replaces the present genotypes. The process is then repeated until the optimal genotype appears. These "transitioning" genotypes are most likely located in local optima (of the landscape) and thus act as exploration hubs for the population. Since these peaks cannot be escaped swiftly, an increase of distinct GRNs will be observed (Fig. 3).

## Robustness of gene regulatory network

Robustness to the (phenotypic) effect of mutations has been studied in the framework of GRNs (*Wagner, 1996*), demonstrating that GRNs which reached the phenotypic optimum are less sensitive to mutations-a phenomenon named epigenetic stability. Thus, epigenetic stability was attributed to the evolution of GRNs *via* the selection process. In order to study this phenomenon, We developed a framework inside EvoNET that allows the simulated population to follow multiple trajectories. Specifically, at discrete time-points EvoNET clones the evolving population ('core' population) creating a 'branch' population. Each

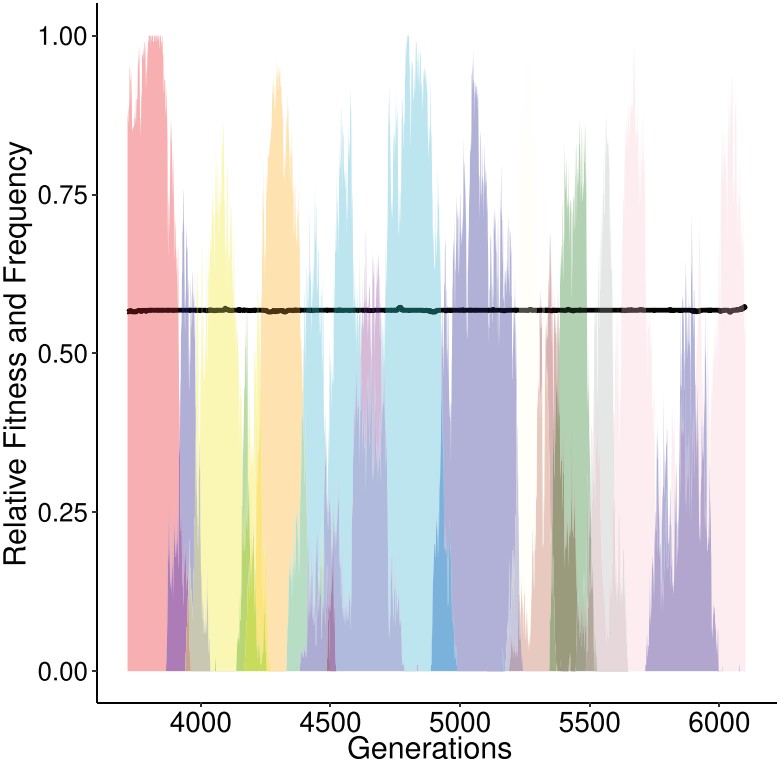

**Figure 3 Alternating frequency-trajectories of the various regulatory networks at a certain fitness level (0.5679; black horizontal line).** Each color represents a distinct GRN haplotype. During this time period the population has a constant fitness (around 0.5679, black line). Here, we show only networks that reach a frequency of at least 50%. There are 14 different networks. This is the result over one simulation that displayed the 'ladder like' behaviour described in Fig. 2.

'core' individual has an interactions matrix $M_i$ shared with its 'clone'. The 'branch' population is subject to a user defined number of mutations and then both populations start the maturation progress. The interaction matrices are then discretized (positive values are transformed to 1, negative to −1 and 0 values remain 0) in order to compare the network topologies of the branch and the core population.

We assess the GRN robustness at two levels, topology and phenotype. Each GRN has a unique network topology characterizing the strength and effect of all gene interactions. In EvoNET, the topologies are modelled by the interaction matrix, so the additional mutations occurring in the 'branch' population have the potential to change the network's topology. Robustness is calculated as the identity between the 'core' and 'branch' interaction matrices after the incorporation of the additional mutations on the 'branch' population. Expression (or phenotypic) robustness measures the identity of the (binary) expression vector between the two populations after every branching (Fig. 4). The robustness of the expression vector is very high in the start of the simulation as the initialization of genotypes does not allow for interactions. Robustness falls dramatically after the initialization step and increases as fitness increases. The maximum robustness is achieved when the optimum has been reached, on average. The topology is less robust than

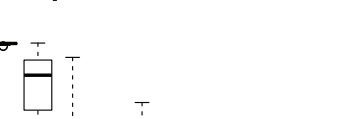

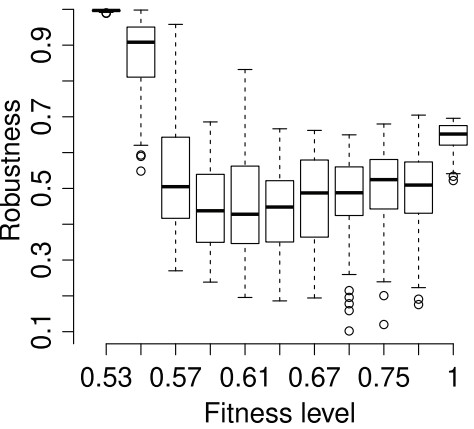
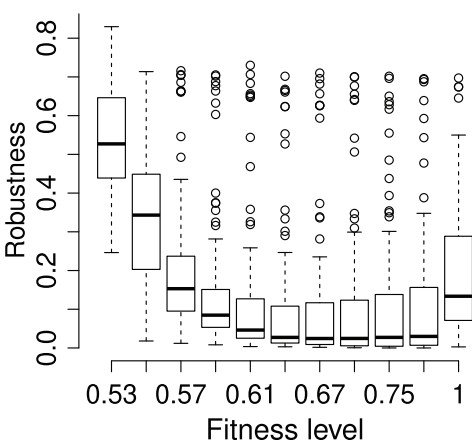

**Figure 4** **Robustness of the (binary) expression vector and network topology.** Each box represents a discrete time point at with the population was split into 'core' and 'branch'. The latter ones were subject to 15 random mutations, followed by the maturation step and fitness calculation. "Expression Robustness" is measured as the identity of the resulting expression matrices, while "Topology obustness" is measured as the identity of the populations' interaction matrices.

then expression vector. However, robustness of topology also increases when the population has reached the maximum fitness level.

## Effect of neutral genes

All genes in a GRN are not subject to the same evolutionary pressure. Often, a subset of the GRN is evolving under neutrality while other parts are under selection. In EvoNET a gene is under selection if its state directly affects the fitness of the individual (*i.e.*, the fitness is different if the gene is active or inactive). In contrast, the state of a neutral gene does not directly affect the fitness. It might affect the expression of a "selected" gene, thus having an indirect effect on fitness. We calculated that the number of interactions between neutrally evolving genes and selected genes increase, until the population reaches the optimum (Fig. 5). While fitness increases, there are multiple interactions between the two parts (neutral and selected), due to the fact that a mutation in the neutral part of the GRN may have an indirect positive effect on the GRN, probably because it regulates the genes of the GRN that are under selection. In contrast, when the population is at the optimum (Fig. 5, right box), mutations are rather deleterious resulting in disadvantageous interactions. Since mutations happen with the same rate across both the neutral and selected part of the GRN, the greater the GRN, the larger the probability of deleterious mutations. Thus, interactions that can be eliminated are eventually discarded (Fig. 5).

## Mutational buffering

In traditional evolutionary theory mutations are often modeled to have a set effect on individuals' fitness. In a model with regulation, the relationship between genotype and fitness becomes considerably obscure. On one hand, mutations on "neutrally" evolving genes may change the regulation of genes that affect the phenotype, thus having an indirect

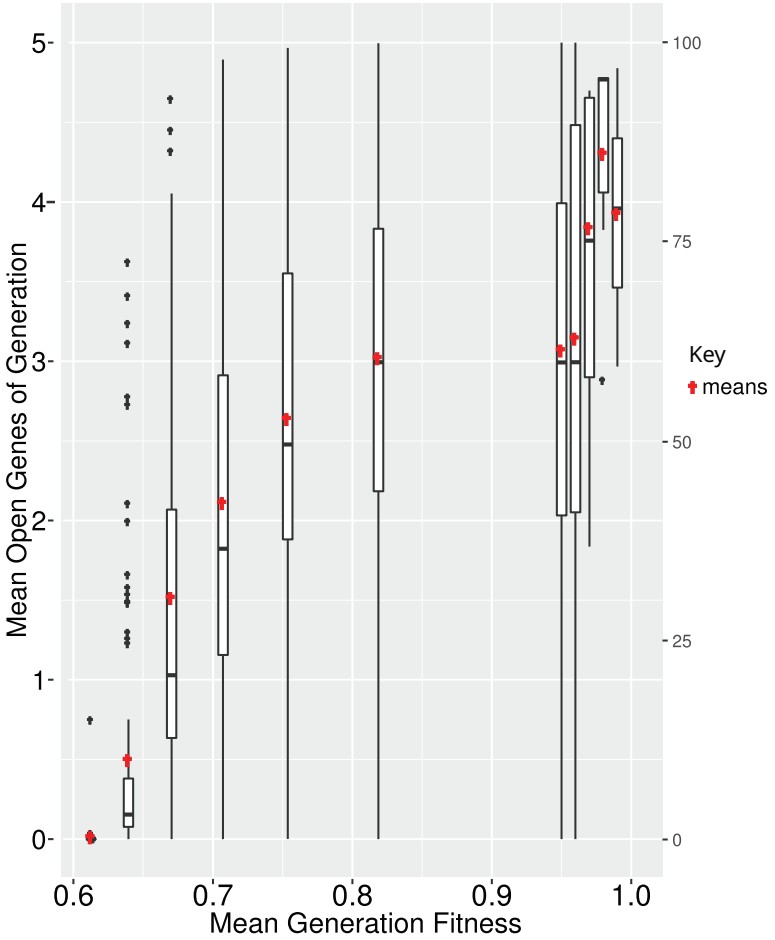

**Figure 5 It is beneficial for the GRN to interact with neutrally evolving genes when the population is ascending the fitness landscape (boxes; red points represent the means).** Upon reaching optimum fitness those interactions tend to be discarded. Boxplots depict averages of 100 simulations, where the majority reached each fitness step.

effect on fitness. In contrast, mutations on "selected" genes may not change the overall phenotype, thus having no effect in fitness (*Krishnan, Tomita & Giuliani, 2008*).

In order to access the role this effect has on the time that optimal fitness is reached, we compared EvoNET with a simpler algorithm that omits the GRN and directly switches the expression of genes on and off. We demonstrate that the existence of the GRN gives rise to mutational robustness and therefore reaching the fitness optimum faster at high mutation rates. We observe that as mutation rate increases, the two strategies display different behaviour (Fig. 6). For small mutation rates the fitness optimum is reached substantially slower by the GRN because robustness and the resulting buffering of the mutation effects hinders the traversal of the fitness landscape. When the mutational load increases, however, the traditional model shows a sharp increase in the time required to reach the optimum. Individuals that have reached a higher fitness will pass potentially different genomes to their offspring. This effect is mitigated in the case of GRNs because of their robustness (Fig. 6).

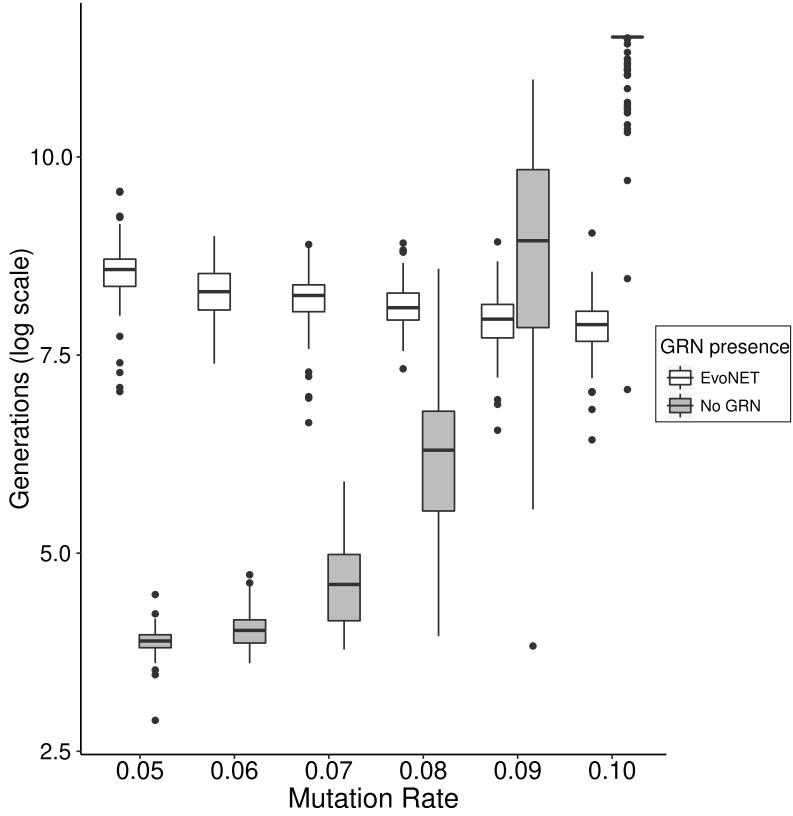

**Figure 6 Comparison between the time (in generations) needed to reach the fitness optimum between EvoNET (white) and a simple model with a non-interactive set of genes.** For lower mutation rates, the model without interactions needs less time to reach the fitness optimum. The opposite trend is observed for high mutation rate values. A total of 100 simulations were performed for each mutation rate value and each scenario.

## DISCUSSION

In recent years, we have witnessed progress on the discovery of GRNs, especially *cis*-regulatory modules (CRMs). In addition, with the assistance of machine learning tools, the importance of GRNs in our understanding of phenotype formation has been highlighted (*Kantorovitz, Robinson & Sinha, 2007*; *Kantorovitz et al., 2009*; *Kazemian et al., 2011*). There is a gap though, in our understanding of the effect of the biological organization (from genes to GRNs and eventually to phenotypes) on the fitness of individuals.

For this reason we created EvoNET. EvoNET creates a detailed model of regulation of a phenotype by implementing and extending Wagner's model of regulation. By implementing *cis* and *trans* regulatory regions as part of the network, we are able to simulate populations and link their individuals' GRNs with a fitness effect. We also offer considerable improvement upon previous models by implementing a more realistic recombination model and also by not discarding but handling cyclic equilibria in the maturation process, contrary to previous studies. We introduced a new recombination model (R1R2) that is more realistic than the previously used row-swapping model by *Wagner (1996)*. The R1R2 model has a similar behaviour with Wagner's row swapping model regarding the average time needed for every fitness level (Fig. S4).

As with any simulation study, it is imperative to acknowledge and address a series of underlying assumptions inherent in its developmental stages. A pivotal decision in this regard was the discretization of expression vectors, where the phenotype of a gene is divided into either expressed or non-expressed states. While this binary representation simplifies the computational framework, it disregards the nuanced and quantitative expression patterns observed in certain biological networks. The rationale behind this approach was to streamline the maturation process computationally. Furthermore, a noteworthy constraint lies in the model of interaction between *cis* and *trans* regions. The current implementation assumes an equal amount of interaction points within these regions, exclusively influenced by their individual states. Consequently, this framework precludes the consideration of non-genetic factors, such as methylation events, in shaping phenotypic outcomes. In addition, aligning simulation parameters, such as mutation and recombination rates, with empirically derived values becomes ambiguous, given the inherent simplifications in the model. Another possible point of scrutiny is also our decision to simulate haploid individuals. *Wagner (1996)* does provide some insight that informed such decision: "It is not clear *a priori* whether diploidy would further increase the magnitude of the effects observed here, because evolution of specific dominance relations among alleles seems possible in a model like this. However, it is unlikely that diploidy would diminish these effects." Apart from dominance effects, incorporating diploidy would add only double the number of genes, but the number of interactions would increase exponentially skyrocketing the computational cost of EvoNET.

In our simulations, with moderate values of mutation and recombination rate, the exploration of fitness landscape follows a ladder-like behaviour, implying that adequate amount of time is needed until certain mutations will bring the population to the next fitness level (Fig. 2). At first glance, this observation may point to a saltatory model of evolution. Saltatory evolution (SE) hypothesizes periods of rapid increase in mutation rate—often linked with the development of beneficial traits (*Theißen, 2009*). What we observe in simulations that display this ladder-like change in population fitness however, is that variability doesn't lead into rapid changes of populations' fitness (Fig. 3) but if such a change occurs, the population quickly adopts this 'fitter' genotype. During this 'adoption phase' it is safe to assume that variability in the population will drop and will steadily increase as the populations traverses the next 'step' of the ladder. In contrast, SE suggests that most of the variability will be generated rapidly and lead to an increase in the development of beneficial traits. Moreover, since EvoNET does not support intermediate expression levels, shifting the state of a gene towards the phenotypic optimum will cause a jump in fitness that might imply saltation. Conclusively, we believe that our results cannot be received as evidence for the SE model, firstly because in the phenotypic level, the binary-expression model forces a jump and secondly, on the genotypic level, we do not observe a saltation.

We explored the role of robustness of the GRNs while they undergo selection. Robustness implies the existence of phenotypically neutral mutations and allows for complex biological structures that are resistant to the detrimental effects of mutations.

There are two layers that provide robustness to the network, the network topology and the phenotype. The phenotype is more robust to mutations than network topologies, since topology is directly related to the regions affected by mutations. By comparing EvoNET with a GRN-less simulation (Fig. 6) we conclude that these robustness layers permit the GRN to increase its fitness even under high mutation rate. In lower mutation rates, robustness acts as a barrier on the effect of all mutations driving the population to a flat network space thus avoiding perturbations (*Lenski, Barrick & Ofria, 2006*). In contrast, when the mutation rate increases, the GRN robustness limit is overcome and deleterious mutations, eventually affect the fitness of the population. Thus, at least up to some threshold, GRNs are able to buffer the detrimental effect of mutations, highlighting their biological significance.

A similar phenomenon was also noted by *Wagner (1996)*, who postulates that although certain states may exhibit equivalent fitness levels, natural selection could operate indirectly. He theorizes that if there exist gene regulatory networks within populations whose mutants consistently yield lower fitness, such networks would gradually be phased out through selective pressure. This proposition, however, arises inquiries into the nuanced understanding of fitness, not solely within the realm of computational simulations but more expansively, as a biological attribute of organisms. Consider a scenario wherein two individuals exhibit identical rates of reproductive success, yet their offspring consistently vary in fitness owing to the susceptibility of their genetic makeup to mutations. Were fitness interpreted solely as the reproductive likelihood of individuals (as is implemented in EvoNET), it would appear that these individuals possess equivalent fitness; nonetheless, it is evident that one genome would substantially outperform the other in the long term. This scenario underscores the intricacies of fitness determination and prompts exploration into the heritability of fitness traits. Furthermore, it beckons the investigation of whether such phenomena are inherent components of biological processes or mere artifacts resulting from the constraints imposed by simulation frameworks. Resolving these inquiries is pivotal for a comprehensive understanding of the interplay between genotype, phenotype, and evolutionary dynamics.

In EvoNET we can allow for genes that do not affect the fitness of an individual directly (neutral genes); however, they may interact with genes that directly affect fitness. These dispensable genes, which are not critical for an organism's basic survival but may provide benefits under certain conditions, can play a useful role in steering a population towards an optimal adaptation more swiftly. The main benefit of having dispensable genes is their role in adaptive flexibility. Thus, a hypothesis that needs to be tested more thoroughly and our simulations provide evidence for its validity (Fig. 5), is that dispensable genes may help populations climb adaptive peaks faster by offering multiple genetic pathways to explore and exploit, speeding up the evolutionary process and helping organisms adapt more quickly than they might with a less diverse genetic toolkit. In addition, when the population is very close or has reached the optimum, we observe a reduction in gene interactions. Dispensable genes introduce a layer of genetic diversity that can be especially advantageous when environmental or even genomic conditions change. In a stable

environment, these genes might remain neutral, not providing any significant advantage or disadvantage. However, when conditions shift these genes can suddenly become beneficial. A study by *Gerdol et al. (2020)*, suggested that in mussels, dispensable genes usually belong to young and recently expanded gene families enriched in survival functions, which might be the key to explain the resilience and invasiveness of this species.

## CONCLUSIONS

Gene regulatory networks play an intermediate role between the genotype and the phenotype. In order to study their role on the evolution of populations, we developed EvoNET, a versatile simulator for the evolution of GRNs through means of genetic drift and selection. We improved upon previous models of recombination and introduced a novel method for dealing with cyclic equilibria. Thus, we were able to demonstrate the effects of GRNs on the genetic robustness as populations traverse the fitness landscape, as well as verify previous findings. Lastly we discuss a series of limitation that underlying model assumptions impose and provide areas that require further understanding. The source code for EvoNET can be found at https://doi.org/10.5281/zenodo.11215048.

### Funding

This work was supported by an internal grant of ICS-FORTH to Pavlos Pavlidis (Grant: ESO00121, EVONMDA). There was no additional external funding received for this study. The funders had no role in study design, data collection and analysis, decision to publish, or preparation of the manuscript.

### Grant Disclosures

The following grant information was disclosed by the authors:
Internal grant of ICS-FORTH to Pavlos Pavlidis: ESO00121, EVONMDA.

### Competing Interests

The authors declare that they have no competing interests.

### Author Contributions

- Stefanos Papadadonakis performed the experiments, analyzed the data, prepared figures and/or tables, authored or reviewed drafts of the article, and approved the final draft.
- Antonios Kioukis conceived and designed the experiments, analyzed the data, prepared figures and/or tables, authored or reviewed drafts of the article, and approved the final draft.
- Charikleia Karageorgiou performed the experiments, analyzed the data, authored or reviewed drafts of the article, and approved the final draft.
- Pavlos Pavlidis conceived and designed the experiments, analyzed the data, prepared figures and/or tables, authored or reviewed drafts of the article, and approved the final draft.

# PeerJ

## Data Availability

Data and code are available at Zenodo:

evolabics, & Stefanos_Papadantonakis. (2024). evolabics/evonet: EvoNET v1.0.2 (1.0.2). Zenodo. https://doi.org/10.5281/zenodo.11215048.

## Supplemental Information

Supplemental information for this article can be found online at http://dx.doi.org/10.7717/peerj.17918#supplemental-information.

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
