# Peer review of "Evolution of gene regulatory networks by means of selection and random genetic drift"

_PeerJ, doi:10.7717/peerj.17918_

## Round 0.1 · original submission · Major Revisions

I encourage the authors to restructure the manuscript for clarity as suggested by reviewer 2 and include the points raised by reviewers 1 (the relevance of GRN to study selective sweeps) and 2 (gradual vs. saltatory evolution) in the introduction and discussion sections respectively.

·

Basic reporting

The manuscript by Papadantonakis and colleagues present a very interesting and well-thought novel approach to investigate the evolution of gene regulatory networks (GRN) using a simulation framework that they developed at this purpose. The general model of Wagner and other subsequent researchers on GRN is improved and I think that the new implementations in the model provide a valuable contribution in the study of genotype-phenotype evolution.
I provide some and specific general comments that the authors may consider to improve their manuscript.

Experimental design

no comment

Validity of the findings

no comment

Additional comments

General comments:
In the introduction the authors spend some time explain why GRN are relevant when studying selective sweeps, but it is not clear why they are so relevant to the problem. For example, I suggest explaining the rationale behind the critique outlined in lines 37-39; selective sweeps are by definition associated to selection at a particular locus, and therefore are expected to occur independently on whether such locus is at the basis of a phenotypic trait (sensu latu) that is under the control of more loci (including their interactions). The following paragraphs explains some of it, but in my view should be introduced more clearly, e.g. by distinguishing dynamics and effects of such processes (I hope I have made myself clear).
In your model E is a binary vector, indicating only if a gene is active or not: can you later discuss the limitations of such assumption? The strength of the interaction may in fact be biologically relevant as it may correspond to more stable and hence efficient gene regulation.
I think that Discussion reads more as a recap than a reasoned discussion of the results. I suggest to just focus on the improvements of EvoNET compared to the previous models and highlight possible limitations or additional future steps. I also wonder whether it would make sense to discuss how the notion of “dispensable genes” (see e.g. https://doi.org/10.1038/35082561, https://doi.org/ 10.1101/gr.87702, https://doi.org/10.1098/rsbl.2008.0732, https://doi.org/10.26508/lsa.202302192) may be related to GRN in your study, as well how the evolution of alternative phenotypes (including in phenotypic plasticity) can be associated to the fraction of “neutral” and “selected” genes, which may then favour exploring alternative fitness peaks, including when environment is unstable (e.g. https://doi.org/10.1086/714530, https://doi.org/10.1016/j.biosystems.2022.104791 , https://doi.org/10.1111/jeb.13527, https://doi.org/10.1073/pnas.1104825108).

Specific comments:
Lines 33-34: i would introduce separately simple environmental effects and phenotypic plasticity.
Lines 68-69: I think that the authors should better explain how such modification of gene expression is related to gene networks. GRN are defined in the abstract, please define them in the main text as well.
Line 70: Mutation rate, if present, is by definition >0, hence not clear why this is stated
Line 88: change from “a quantitative trait loci” to either “quantitative trait loci” or “a quantitative trait locus”
Line 99: I would be more explicit on which differences there are.
Paragraph starting at line 104: I think that you should better define what you mean by “trans region of gene j”, as trans typically refers to elements/factors that bind cis elements, and if I understand correctly, should correspond to the binding portion of the gene j itself. In fact, I think that you should provide a simple explanation of what L should be (i.e. ~length of the interaction between cis and trans elements);
Paragraph starting at line 104: the [-1,1] interval includes both strength and kind of interaction
Line 118: The matrix assumes that individuals are haploid, please specify that
Line 146: delete “bothincrease”
Line 168: any justification for the choice of this very specific value? Have you tested other values, e.g. 0.1% or 10%?
Line 172: the “optimal vector of expression” should be better defined: how do you define “optimal”? Can it correspond to the average population value, so to allow the estimation of a relative fitness for each individual?
Equation 3: please define sigma2
Line 176: (two lines down): not clear what “constant” means in this context: same E for all? How is E determined?
Line 185: I suggest changing to “individuals in which cyclic equilibria are reached” or similar
Line 194: this section is more pertinent to the M&M
Line 209: change “person” to “individual”
Line 212: can you discuss the possible differences that a model with diploid individuals will give? (apart from the obvious added complexity due to dominance effects)
Line 222: given the defined initial and optimal states, isn’t it expected to have such steps corresponding to a single gene activation?
Line 226-227: do you mean that the average population fitness also increases?
Line 245: I would add that the distinct GRNs are to be intended over time.
Figure 4: I have some problems in understand the figure, as the described this effect would be more evident if time (generations) was put in the X axis: what does it mean having Generations=0 between Fitness Steps 2.5 and ~5? How do you define fitness steps?
Lines 265-268: Are the “complementary” 0000011111, 0011001100 and 0101010101 behaving in the same way?
Lines 279-280: there seem not to be a section dedicated to the explanation about the interaction matrices.
Lines 287-288: how is robustness calculated? (see also Fig 7)
Line 289: remove
Figure 8: this reminds me of (source-sink) meta-population dynamics, could it make sense?
Line 305: do you foresee the possibility that selection may act to reduce the number of interacting genes?
Line 335: change “netwo.0, Emacs rk” to “network”
Lines 340-346: redundant with previous paragraph

Reviewer 2 ·

Basic reporting

In this study, Papadadonakis et al correctly highlight that gene interactions have been overlooked when studying gene evolution under selection and neutrality and that this order of organization has to be taken into account moving forward. They use evolutionary simulations based on a modified version of a well-known gene regulatory network model to address this question. The modifications to the Wagner GRN model are minor, although the implementation of recombination through distinct cis and trans regulatory regions the authors have devised is an improvement to original model. The biggest merit of this paper is the application of the model, which is obfuscated by the manuscript structure and writing. I think this paper has merit, but it needs large restructuring before it can be published. I list some points which need to be addressed below.

Major:
- Many main figures are not relaying important information and can be moved to the SI. Figure 1 is a diagram explaining the recombination scheme of the model, which provides a clearer explanation of recombination model described in the main text. However, I don’t see what information the reader is given in Figures 2-6, which depict some aspect of the evolutionary simulation without a clear point. For example, Figure 3 shows new mutant network genotypes rising in frequency and then being replaced by more fit mutants. What does the reader learn from this plot, other than learning that the population evolved by fit individuals rising in frequency in the population? Figure 4 depicts a positive correlation between the mutation rate and speed of evolution, a common fact used in in-silico evolution experiments to speed up evolution and conserve computation time. The fact that these figures corroborate what is already known is a very good diagnostic that the evolutionary simulation framework is working well, but offers no new insight to the reader. I recommend this part of the paper to be summarized and figures compiled as subplots into a single figure or moved to the SI, so that other figures (7-9), with novel results, can be highlighted.

- Section 3.1.2 “Optimum is gradually reached in a ladder-like fashion” implies a finding which seemingly adds evidence to the gradual vs. saltatory evolution debate, i.e. does evolution in nature proceed in slow and gradual changes or in large ladder-like jumps, which are preceded by periods of stagnation. However, the experimental design of the model necessarily imposes a ladder-like evolution dynamic. Namely, the gene expression is binary (0 or 1) and the imposed optimum is the phenotype of all genes being on. Whenever the interactions in the network mutate sufficiently to switch the gene from a 0 to 1, the fitness jump will happen, because there are no intermediate phenotypes (gene expression being 0.5, for example), and the average fitness of the population steps up one step on the “ladder”. I would be careful drawing conclusions here and mentioning it again in the discussion, as the experimental design strongly favors saltatory evolution. As such, the only information given by this paragraph is that the populations they put under selection managed to respond to selection, which is a validation of their evolutionary framework, but nothing novel.

- Section 3.2 “Interplay between Recombination and Optimum state” reports that recombination has a different effect in populations under different fitness optima on a sample of 4 optima. I think this is insufficient sample size to make any conclusion, which the authors point out, but include as a section nonetheless.

- Writing - I suggest another detailed read of the manuscript because there are typos and mistakes such as started sentences (line 289) or misplaced copy-pasted content (line 335). When referencing conclusions from previous work by Andreas Wagner, it is not necessary to specifically refer to figures in the paper. The introduction can be simplified and streamlined, e.g. by removing some sentences (I made some suggestions in the attached pdf). Also, a big part of the introduction describes selective sweeps and the methods used to infer them, but selective sweeps are not referenced anywhere later in the manuscript.

Minor:
- Section 3.1.3 “Size of the regulatory space in neutrality and selection” reports that there was a higher number of distinct genotypes overall throughout the evolutionary simulation in populations evolved under selection than under neutrality. I find it hard to believe that a population under any selective constraint would cover more area on a fitness landscape than a population randomly drifting through with no constraints. Could you please provide some more information (like how is neutrality implemented) and explanation for this result?

Experimental design

- How many simulations were performed in each scenario, was there only a single evolutionary run or multiple with different initial conditions for each scenario? The figures do not include the number of evolutionary runs/sample size, only summarized information.
- How is evolution under neutrality implemented, is there a fitness calculation step? It is not reported anywhere in the maintext, except noting a “binary flag that denotes whether simulation is neutral” (line 198).
- Section 3.3: How is robustness defined? It is introduced as being a focus of previous research by Wagner, but not defined in the manuscript.

Validity of the findings

Source code is provided.

Annotated reviews are not available for download in order to protect the identity of reviewers who chose to remain anonymous.

---

## Round 0.2 · accepted · Accept

I thank the reviewers for the thorough revision and the manuscript is much improved now. Both reviewers recommended accepting the paper, which I concur.

·

Basic reporting

No comment

Experimental design

No comment

Validity of the findings

No comment

Additional comments

I have read the revised version of the manuscript and the responses to the comments I provided in the first version. I am fully satisfied with the answers and modifications.

Reviewer 2 ·

Basic reporting

The authors have greatly improved the clarity of the manuscript and addressed all of my concerns. Explanations for unclear terms have been added, the results and figures have been streamlined, figure captions refined, the discussion expanded, and the overall writing improved. The scientific question and novelty of the findings in this paper is more clear to the readers, and limitations to the model have also been added to the discussion, which I appreciate. I commend the authors and recommend this manuscript for publication.

Experimental design

No comment.

Validity of the findings

No comment.